# Cycle Threshold (Ct) Values of SARS-CoV-2 Detected with the GeneXpert® System and a Mutation Associated with Different Target Gene Failure

Keita Yamashita *, Terumi Taniguchi, Noriyasu Niizeki, Yuki Nagao, Akira Suzuki, Akihiro Toguchi, Shiori Takebayashi, Jinko Ishikawa, Osanori Nagura, Kazuki Furuhashi, Moriya Iwaizumi and Masato Maekawa *

Department of Laboratory Medicine, Hamamatsu University School of Medicine, Hamamatsu 431-3192, Japan
* Correspondence: keitay@hama-med.ac.jp (K.Y.); mmaekawa@hama-med.ac.jp (M.M.);
  Tel.: +81-53-435-2723 (K.Y.); +81-53-435-2721 (M.M.)

**Abstract:** SARS-CoV-2 nucleic acid detection tests enable rapid virus detection; however, it is challenging to identify genotypes to comprehend the local epidemiology and infection routes in real-time qRT-PCR. At the end of June 2022, our hospital experienced an in-hospital cluster of COVID-19. When examined using the GeneXpert® System, the cycle threshold (Ct) value of the N2 region of the nucleocapsid gene of SARS-CoV-2 was approximately 10 cycles higher than that of the envelope gene. Sanger sequencing revealed a G29179T mutation in the primer and probe binding sites. A review of past test results revealed differences in Ct values in 21 of 345 SARS-CoV-2-positive patients, of which 17 cases were cluster-related and 4 were not. Including these 21 cases, 36 cases in total were selected for whole-genome sequencing (WGS). The viral genomes in the cluster-related cases were identified as BA.2.10, and those in the non-cluster cases were closely related and classified as being downstream of BA.2.10 and other lineages. Although WGS can provide comprehensive information, its use is limited in various laboratory settings. A measurement platform reporting and comparing Ct values of different target genes can improve test accuracy, enhance our understanding of infection spread, and be applied to the quality control of reagents.

**Keywords:** SARS-CoV-2; GeneXpert; Ct value; G29179T; infection control; quality control





## 1. Introduction

Nucleic acid detection tests are used to diagnose severe acute respiratory syndrome coronavirus 2 (SARS-CoV-2) infection, and several devices and reagent kits are commercially available for this purpose [1–4]. Among these, real-time reverse transcription polymerase chain reaction (qRT-PCR) assays have formed the backbone of coronavirus disease-19 (COVID-19) diagnosis since the start of the COVID-19 pandemic, and many qRT-PCR-based SARS-CoV-2 testing platforms have been developed and made commercially available, including the GeneXpert® System and Cepheid Xpert Xpress SARS-CoV-2 assay kit (GX) (Beckman Coulter, Inc., Brea, CA, USA) [5], which target the envelope gene (E) and N2 region of the nucleocapsid gene (N2) of SARS-CoV-2. If a sample is positive, it is expected that N2 or both N2 and E targets will be detected using threshold (Ct) values [6].

In our hospital, various SARS-CoV-2 nucleic acid detection tests are used. Particularly, GX is used for routine and emergency testing. We identified a cluster of COVID-19 cases within a short period among hospitalized patients and staff members from June to July 2022. Suspecting a nosocomial infection, we used GX to screen many contacts, and some SARS-CoV-2-positive cases were detected among individuals who had not yet developed symptoms.

Whole-genome sequencing (WGS) of the viral genome can help identify the virus strains and can be useful for clarifying whether these cases originated from a single infection route; however, it is difficult to perform WGS in hospital laboratories owing to the time, cost,

equipment, and competence required. We noticed a characteristic preliminary finding in the GX results—the two Ct values of the N2 and E gene targets were markedly different: the Ct value of N2 was approximately 10 cycles higher than that of E. Mutations in the qRT-PCR primer and probe binding sites have been reported to cause the discrepancy in the Ct values of these two gene targets [7–22]; therefore, we aimed to determine the base sequence of the GX detection site. In addition, WGS was performed using next-generation sequencing. We also traced the GX results to determine when viruses with these characteristics first emerged. From June to July 2022, when the cluster of COVID-19 cases occurred in our hospital, the BA.2 lineage of the Omicron variant was replaced by the BA.5 lineage, and both the BA.2 and BA.5 lineages were identified in Japan. In this study, we verified whether SARS-CoV-2 point mutations can be detected by real-time qRT-PCR using Ct values and tested the validity of this tool for estimating nosocomial infection routes. This study is a coincidental product of the cluster analysis of cases concentrated in the same department within a specific period. The cases were believed to be a cluster based on evidence (several occurrences in the same department at the same time).

## 2. Materials and Methods

### 2.1. Participants

The study participants included 36 patients and staff members who tested positive for SARS-CoV-2 through qRT-PCR from June to early August 2022. We also reviewed the GX test results of 345 positive cases out of 4905 patients tested for SARS-CoV-2 at Hamamatsu University Hospital from mid-June 2020, when COVID-19 was first confirmed in Hamamatsu, Japan, to early August 2022.

### 2.2. Nucleic Acid Detection Testing of SARS-CoV-2

Nasopharyngeal swab samples were collected from selected patients with a fever, sore throat, headache, and fatigue and from persons they had had close contact with, including the medical staff of Hamamatsu University Hospital (Hamamatsu, Japan). Flocked nasopharyngeal swabs were placed in a BD Universal Viral Transport Collection Kit (Becton, Dickinson and Company, Franklin Lakes, NJ, USA) and transported to the clinical laboratory of the hospital. qRT-PCR for SARS-CoV-2 was performed using the GeneXpert® System and Cepheid Xpert Xpress SARS-CoV-2 assay kit (Beckman Coulter, Inc., Brea, CA, USA), according to the manufacturer's protocol. The GeneXpert® system is a fully automated genetic analyzer that integrates the processes of nucleic acid extraction, PCR amplification, and detection. In addition to SARS-CoV-2, it also supports microbiological tests, such as those for *MRSA* and *Clostridium difficile*, and is characterized by the rapid reporting of results within 60 min. The Cepheid Xpert Xpress SARS-CoV-2 assay kit employs a cartridge-based nucleic acid amplification test and requires no special treatment by simply applying the sample to the reagent cartridge. The test reports a positive result when N2 is detected, whereas a presumptive positive result is reported when only the E target is detected. The cutoff Ct value for SARS-CoV-2 detection was set at 45, and that for undetectable cases, in which GX failed to yield Ct values, was set at 0. As a quality control, a sample processing control (SPC), which functions as an internal control, is included in the cartridge of the Cepheid Xpert Xpress SARS-CoV-2 assay kit [6].

### 2.3. DNA Sequencing of the SARS-CoV-2 Genome

Total RNA was automatically extracted from nasopharyngeal swab samples using the QIAcube® platform (Qiagen, Venlo, The Netherlands), and 5 µL of the eluate was used to prepare cDNA using the ReverTra Ace-α-® qPCR RT Kit (Toyobo, Tsuruga, Japan). Sanger sequencing was performed targeting the N2 region of SARS-CoV-2 using a BigDye™ Terminator v3.1 Cycle Sequencing Kit and an Applied Biosystems SeqStudio Genetic Analyzer (Thermo Fisher Scientific, Waltham, MA, USA). The primers used are listed in Table S1 [23,24].

WGS of SARS-CoV-2 was performed using targeted amplification of the SARS-CoV-2 genome by multiplex PCR according to the National Institute of Infectious Diseases (NIID, Tokyo, Japan) protocol [25] using the MiSeq platform (Illumina, San Diego, CA, USA). The SARS-CoV-2 genome library for the MiSeq platform was prepared using the Super-Script IV First-Strand Synthesis System (Thermo Fisher Scientific), Q5 Hot Start DNA Polymerase (NEB), QIAseq FX DNA Library UDI Kit (QIAGEN), and Alt nCov2019 primers/Primers/ver N5 [26].

We identified the virus strains of the selected 36 infected individuals through WGS. The obtained sequence reads were compared to those of the reference strain (Wuhan-Hu-1, NC_045512) using the SARS-CoV-2 pipeline (https://github.com/onecodex/sars-cov-2 (accessed on 28 February 2023)), and consensus sequences of each sample were generated using iVar (https://andersen-lab.github.io/ivar/html/index.html (accessed on 28 February 2023)) [27]. The sequence reads of three patient samples could not be obtained owing to the low quality and number of reads. A total of 33 patient sequences in FASTA format were analyzed using "Nextclade (version 2.9.0)" (https://nextstrain.org/nextclade/sars-cov-2 (accessed on 28 February 2023)).

*2.4. Statistical Analysis*

All statistical analyses were performed using 'EZR' (Easy R) [28]. The data are presented as the median and interquartile range (IQR). Differences in Ct values between N2 and E were compared based on c.29179G > T using the Mann–Whitney U test. The significance level was set at $p < 0.05$.

## 3. Results

*3.1. N2 Delay in GX*

Thirty-six patients and medical staff in the same ward or central medical support department suspected of being infected with SARS-CoV-2 were subjected to SARS-CoV-2 screening using GX. An unusual delay or failure in the amplification of N2 against E was observed in 19 cases within 10 days. The Ct values (mean ± standard deviation) of positive cases with the unusual delay (23 cases) were 22.4 ± 6.2 cycles for E and 33.0 ± 7.0 cycles for N2. The Ct delay, calculated as ΔCt (N2—E), was 10.7 ± 2.3 cycles (mean ± SD). Two samples showed an N2 dropout, but the Ct values of E in the samples were higher than 30 cycles, leading to an N2 dropout by a delay of approximately 10 cycles (Table 1; pink group). In the case of numbers 23, 25–27, and 29–35, there was a high amount of virus in the sample (Ct values < 20.0 cycles), dNTPs were consumed, and the SPC could not be amplified (Table 1; green group).

**Table 1.** Details of the 36 selected COVID-19 cases, including Ct values and DNA sequencing data.

| No. | Day | Sex | Age | Patient or Staff | Ct Value (GeneXpert) | | | | Sanger Sequence of N2 | | NGS | Pangolin Lineage | Nextclade Lineage |
|---|---|---|---|---|---|---|---|---|---|---|---|---|---|
| | | | | | E | N2 | SPC | N2-E | c.29179 | Other Mutations | c.29179 | | |
| 1 | −36 | F | 31 | Outpatient | 15.7 | 25.2 | 29.2 | 9.5 | T | | T | BA.2.10 | 21L |
| 2 | −3 | F | 33 | Outpatient | 19.4 | 29.6 | 28.2 | 10.2 | T | | T | BA.2.10 | 21L |
| 3 | −1 | F | 78 | Outpatient | 20.1 | 22.3 | 27.9 | 2.2 | G | c.28983A > G c.29218C > T | G | BA.2.3.1 | 21L |
| 4 | 0 | F | 11 | Outpatient | 20.3 | 28.7 | 27.8 | 8.4 | T | | T | BA.2.10 | 21L |
| 5 | 0 | M | 65 | Medical engineer | 21.9 | 30.8 | 27.7 | 8.9 | T | | T | BA.2.10 | 21L |
| 6 | 0 | M | 24 | Medical engineer | 28.3 | 37.2 | 30.5 | 8.9 | T | | T | BA.2.10 | 21L |
| 7 | 0 | M | 41 | Medical engineer | 21.8 | 33.0 | 28.3 | 11.2 | T | | T | BA.2.10 | 21L |
| 8 | 0 | F | 57 | Nurse | 25.4 | 41.6 | 28.2 | 16.2 | T | | T | BA.2.10 | 21L |
| 9 | 0 | M | 41 | Inpatient | 14.3 | 24.1 | 27.6 | 9.8 | T | | T | BA.2.10 | 21L |
| 10 | 0 | F | 41 | Nurse | 30.7 | 42.7 | 28.0 | 12.0 | T | | T | BA.2.10 | 21L |
| 11 | 1 | M | 27 | Doctor | 15.8 | 25.4 | 28.7 | 9.6 | T | | T | BA.2.10 | 21L |
| 12 | 1 | F | 59 | Nursing assistant | 25.9 | 36.7 | 28.9 | 10.8 | T | | T | BA.2.10 | 21L |
| 13 | 1 | F | 26 | Nurse | 23.2 | 40.0 | 28.9 | 16.8 | T | | T | BA.2.10 | 21L |
| 14 | 1 | F | 25 | Nurse | 34.1 | 43.8 | 28.5 | 9.7 | Failed | | T | Failed | Failed |
| 15 | 2 | F | 26 | Nurse | 32.6 | 43.5 | 27.9 | 10.9 | Failed | | T | Failed | Failed |
| 16 | 2 | F | 47 | Nurse | 33.8 | 0.0 | 28.9 | −33.8 | Failed | | T | Failed | Failed |
| 17 | 2 | F | 26 | Medical engineer | 31.4 | 0.0 | 28.9 | −31.4 | T | | T | BA.2.10 | 21L |
| 18 | 4 | F | 53 | Inpatient | 12.5 | 21.8 | 27.7 | 9.3 | T | | T | BA.2.10 | 21L |
| 19 | 4 | F | 82 | Family of No. 18 | 28.8 | 38.6 | 28.6 | 9.8 | T | | T | BA.2.10 | 21L |
| 20 | 4 | M | 82 | Inpatient | 14.7 | 24.7 | 28.5 | 10.0 | T | | T | BA.2.10 | 21L |
| 21 | 6 | F | 24 | Nurse | 19.4 | 33.5 | 28.2 | 14.1 | T | | T | BA.2.10 | 21L |
| 22 | 8 | F | 27 | Nurse | 26.0 | 35.9 | 28.4 | 9.9 | T | | T | BA.2.10 | 21L |
| 23 | 10 | F | 25 | Medical engineer | 16.7 | 16.9 | 0.0 | 0.2 | G | | G | BF.22 | 22B |
| 24 | 11 | F | 79 | Family of No. 20 | 18.2 | 27.1 | 29.2 | 8.9 | T | | T | BA.2.10 | 21L |
| 25 | 13 | F | 26 | Nurse | 14.7 | 15.3 | 0.0 | 0.6 | G | | G | BA.2.10 | 21L |
| 26 | 14 | F | 26 | Nurse | 19.5 | 20.5 | 0.0 | 1 | G | | G | BA.2.10 | 21L |
| 27 | 14 | M | 77 | Outpatient | 16.6 | 18.7 | 0.0 | 2.1 | G | c.29253C > T | G | BA.2.3.13 | 21L |
| 28 | 19 | F | 24 | Nurse | 25.4 | 25.4 | 30.4 | 0 | Not tested | | G | BA.5.2 | 22B |
| 29 | 19 | F | 26 | Physical therapist | 16.4 | 16.5 | 0.0 | 0.1 | Not tested | | G | BA.5.2 | 22B |
| 30 | 22 | M | 79 | Inpatient | 16.1 | 18.6 | 0.0 | 2.5 | Not tested | | G | BA.5.2 | 22B |
| 31 | 23 | F | 23 | Nurse | 20.7 | 22.6 | 0.0 | 1.9 | Not tested | | G | BA.5.2 | 22B |
| 32 | 23 | M | 29 | Doctor | 18.0 | 19.2 | 0.0 | 1.2 | Not tested | | G | BF.5 | 22B |
| 33 | 24 | F | 37 | Nurse | 19.6 | 20.5 | 0.0 | 0.9 | Not tested | | G | BF.22 | 22B |
| 34 | 25 | F | 25 | Nurse | 19.4 | 19.9 | 0.0 | 0.5 | Not tested | | G | BF.22 | 22B |
| 35 | 33 | F | 27 | Nurse | 18.1 | 19.5 | 0.0 | 1.4 | G | | G | BA.5.2 | 22B |
| 36 | 36 | F | 79 | Inpatient | 21.0 | 30.0 | 28.9 | 9.0 | T | | T | BA.2.10 | 21L |

Day: The day on which the cluster of COVID-19 cases occurred was set as 0 (late June 2022). SPC: Sample processing control, included in the GeneXpert cartridge. The table coloring is categorized based on the G29179T genotype: Pink: 29179T; light green: 29179G.

### 3.2. N2 Amplification Delay Is Linked to the G29179T Mutation

We suspected that all samples with an N2 amplification delay may contain a mutation in the GX primer or probe region. Therefore, we sequenced the samples with and without the N2 amplification delay. Figure 1 shows the sequencing strategy used to reveal the mutation responsible for the N2 amplification delay and a part of the Sanger sequencing results. All samples with an N2 amplification delay had c.29179G > T (G29179T mutation; p.Pro302Pro synonymous mutation) and the samples without an N2 amplification delay had wild-type 29179G (Figure S1). Additionally, case no. 27 had a c.29253C > T (p.Ser327Leu) mutation, and case no. 3 had c.28983A > G (Lys237Arg) and c.29218C > T (p.Phe315Phe) mutations (Table 1); both cases had wild-type 29179G.

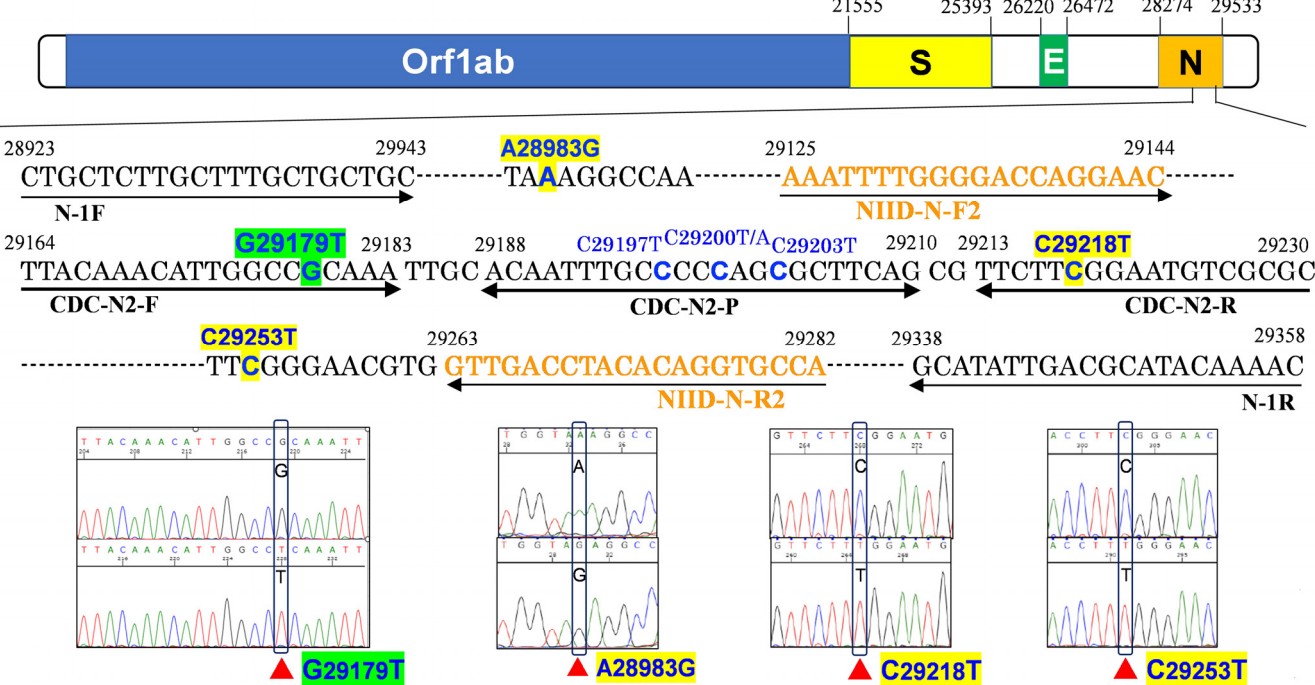

**Figure 1.** Mutations in the nucleocapsid gene of SARS-CoV-2 from patients with COVID-19 and the strategy used to reveal the mutation responsible for the N gene (N2 target region) amplification delay in the GeneXpert® System and Cepheid Xpert Xpress SARS-CoV-2 assay kit (GX) results. Orange bases indicate the N2 target region for Sanger sequencing of the nucleocapsid gene; the genome positions are numbered according to the reference genome (Wuhan-HU-1; NC_045512.2). The primers and probe for the US Centers for Disease Control and Prevention (CDC) primer and probe targets, the Japan National Institute of Infectious Disease (NIID) primers, and the primers designed for the present study are shown. The GX primer targets were very similar to those of the CDC-published PCR primer sets. Green and blue bases indicate the location of the 29179: G/T mutation that spans five nucleotides upstream of the 3' end of the CDC forward primer (possibly the same as the GX primer). Yellow and blue bases indicate other mutation locations: one within the CDC reverse primer (29218: C/T) and two in regions unrelated to any primer or probe (28983: A/G; 29253: C/T). Blue-only bases indicate previously described N gene mutations on the probe binding site that resulted in GX assay failure and instability (29197: C/T; 29200: C/T or A; 29203: C/T).

### 3.3. Past Cases Analyzed through GX

Out of 345 SARS-CoV-2-positive cases, 14 had E2-negative and N2-positive results with high Ct values. In addition, a presumptive positive result was observed in eight cases. Interestingly, only the pink group cases in Table 1 had ΔCt (N2-E) deviations close to 10 cycles for about two years, and 17 cases were cluster-related. Among these, the Ct value for E exceeded 35 cycles and the Ct value for N2 was not obtained in two cases. Because the

Ct values for E exceeded 30 cycles in these cases, it is possible that if the Ct values deviated by approximately 10 cycles, the Ct value of N2 exceeded 45 and became undetectable. For example, in eight cases that have been reported since June 2020, the genetic test results were not available because of the high Ct value (i.e., low viral load). Figure 2 shows the concordance between the Ct values for N2 and E.

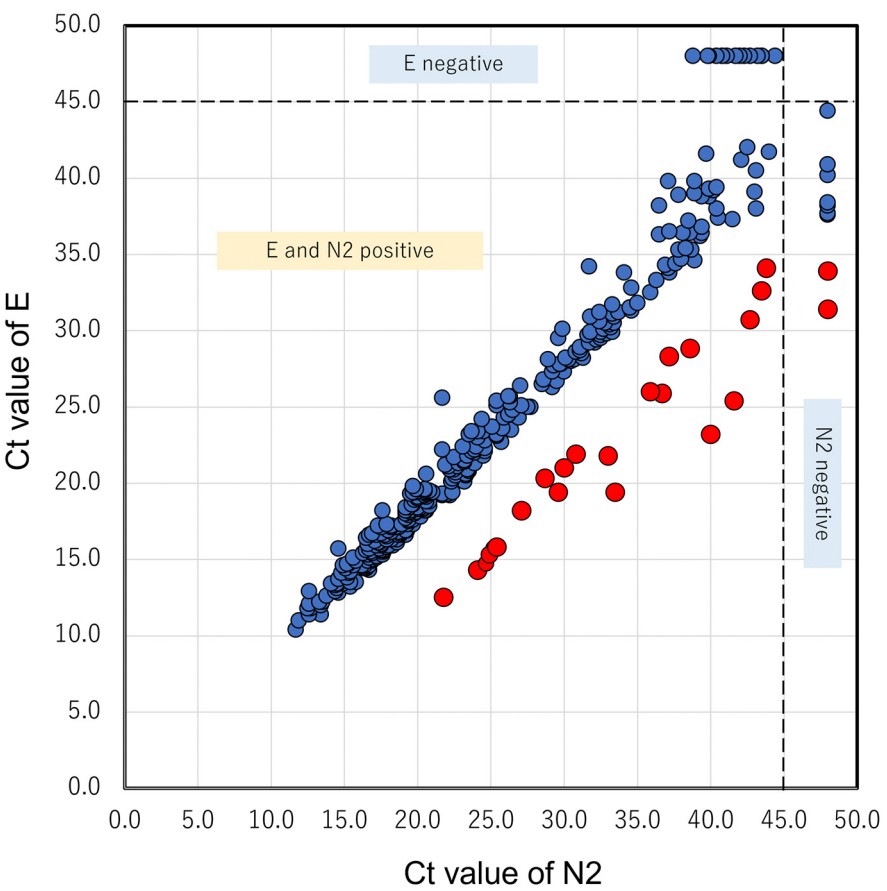

**Figure 2.** Comparison between N2 and E Ct values analyzed through GX. A total of 354 cases with positive or presumptive positive GX results were correlated with two Ct values of N2 and E. Undetectable cases in which GX failed to yield Ct values were plotted with Ct values of 48. Red dots indicate cases with the G29179T mutation.

In the periods preceding and following the cluster, which lasted approximately 10 days, an N2 amplification delay was observed in one outpatient approximately one month before the cluster (no. 1), two outpatients a few days before the cluster (nos. 2 and 4), and one hospitalized patient approximately 1 month after the cluster (no. 36). These four cases did not seem to be related to the cluster. To investigate the relationship between these cases and the cluster, we examined the N gene through Sanger sequencing and found the G29179T mutation in all infected individuals that had an N2 amplification delay.

### 3.4. Phylogeny of SARS-CoV-2 Detected during the Cluster

In 3 of the 36 SARS-CoV-2-positive cases (nos. 14–16) analyzed through WGS (Table 1), the depth and genome quality were too low to confidently assign Pangolin and Nextclade lineages. This was possibly due to an insufficient amount of RNA. For the remaining 33 cases, the genome could be read with sufficient depth and reliability. All 23 cases with an N2 amplification delay and G29179T mutation were classified as Pangolin lineage BA.2.10 and Nextclade lineage 21 L (Omicron) (Figure 3). Of the 23 cases, 19 were cluster-related, 3 (nos. 1, 2, 4) occurred before the cluster, and 1 (no. 36) occurred approximately 1 month after the cluster. There were no similar types of viral infection observed during

the one month. In particular, case no. 1, a woman with familial infection who presented to the emergency department approximately 1 month before the cluster, was identified by reviewing all previous GX results. In addition, one month after the cluster subsided, a similar N2 amplification delay was observed in one case (no. 36). WGS revealed that both of these cases belonged to lineage BA.2.10, with sequences nearly identical to those of the cluster cases. However, slight differences in the base sequence were observed, which was presumed to be due to a subtype of the BA.2.10 lineage that is spreading in the community. In contrast, two cases (nos. 25 and 26) belonging to the BA.2.10 lineage did not have the G29179T mutation. Two other cases belonged to the 21 L (Omicron) BA.2 lineage but to different sublineages—BA.2.3.1 (no. 3) and BA.2.13 (no. 27). The remaining nine cases belonged to the 22 B (Omicron) BA.5 lineage. Among them, three distinct lineages were identified in seven cases (no. 28–34) that worked in a similar environment and developed an infection at almost the same time.

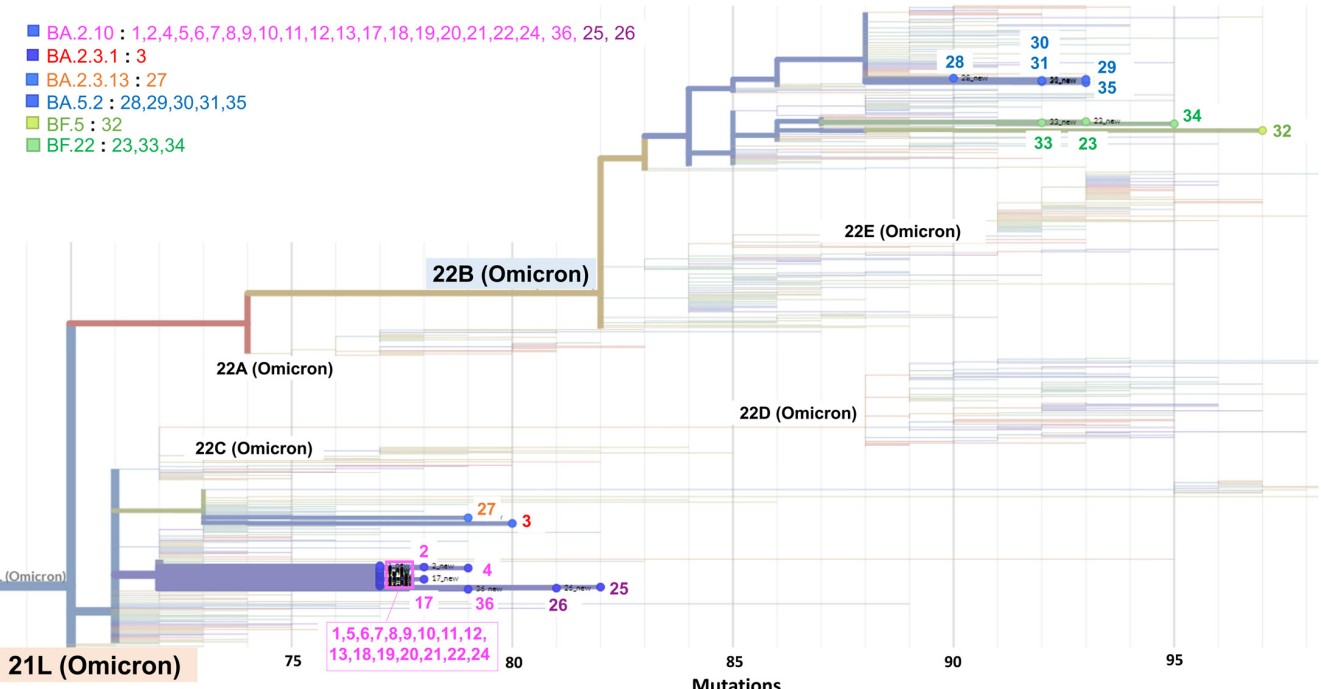

**Figure 3.** Phylogeny of Nextclade and Pangolin lineages. Phylogenetic tree of SARS-CoV-2 showing the distribution of sequences from 33 cases in this study. Each strain classified by Pangolin Lineage is the same color-coded as in Table 1. The numbers also represent the case numbers as listed in Table 1.

## 4. Discussion

In this study, we noticed a discrepancy between the Ct values of N2 and E analyzed through GX, which led to the detection of the G29179T mutation in the PCR primer and probe binding sites. This discrepancy was useful in analyzing multiple cluster cases that occurred at the same time based on changes in the genotype. Therefore, our findings have great significance for viral strain identification and tracking. The diversity of genotypes is one of the most important factors affecting the difference in gene amplification efficiency. Several authors have reported mutations that affect the amplification efficiency of GX [7–22]. Although these mutations are likely to be infrequent, it is important to understand them because they affect the clinical and laboratory assessments of patients with COVID-19. In other words, base substitutions in the binding regions of primers and/or probes are related to the amplification efficiency of PCR, and genotype analysis may lead to the discovery of infection routes and infection control. In this study, cases with the same mutation in the N2 region belonged to the group with the same infection route, and it was inferred that the infection route was different in cases without this mutation, which is of great significance

from the perspective of infection control. Similar to strategies for bacterial infection control in which the genotype of causative bacteria is determined when a nosocomial infection is suspected [29], we demonstrated that it is possible to estimate the route of infection by examining the genotype of viruses, which can be applied beyond the scope of SARS-CoV-2.

A literature search revealed that the G29179T mutation identified herein has already been reported. In Korea, it has a high frequency and accounts for approximately 70% of all mutations in the B.1.497 lineage [7]. In Australia, Foster et al. [8] reported an increase in E-positive and N-negative cases that were caused by a delta strain from August 2021. Many mutations that affect the accuracy of qRT-PCR have been reported to spread to various lineages, including those other than G29179T [9–22]. These mutations are considered to independently occur after being divided into various lineages. In this study, the G29179T mutation occurred in the BA.2.10 lineage and is thought to have spread to many infected people. Furthermore, an individual infected with the same lineage with the G29179T mutation was diagnosed at a different time, suggesting that the strain was present elsewhere. To the best of our knowledge, this is the first report to identify a relationship between the occurrence of the G29179T mutation in the N2 region and the GX delay for the Omicron variant (B.1.1.529 and BA2.10).

From a different perspective, as shown in Figure 2, the relationship between the Ct values of N2 and E is much stronger than expected. This indicates that the amplification efficiencies of the two gene targets maintained a constant relationship; that is, the quality of the reagent in GX was precisely maintained. Conversely, this relationship could be utilized for the quality control of reagents. Many commercially available reagents amplify the two regions as a countermeasure against poor detection (false negative) owing to viral mutations and load [7–22,30–32] or non-specific detection (false positives) owing to cross-reactivity to other viruses [33]. However, some reagents report only positive or negative results without displaying the Ct values, whereas others report only the Ct value of one gene target. The quantitative determination of the Ct values of two or more genes has merits that cannot be obtained from qualitative judgments, such as genotype discrimination and infection control, as shown in this report, as well as the prevention of false positives and false negatives based on these values. The Ct value is a relative, and not an absolute, representation of the viral load and varies according to the measurement method. Nevertheless, if the same method is used consistently and quality control is implemented correctly, the Ct value not only shows the amount of virus in the sample but also partly identifies the genotype. Therefore, it is recommended that the Ct values of all amplified parts be quantified and generated. Our results show that a deviation in the Ct value has great clinical and laboratory significance.

The phylogenetic tree was also used to determine the viral genome, and although it provides crucial information both in terms of infection control and epidemiology, it remains necessary to perform WGS [34,35]. However, because it is difficult to perform WGS for routine laboratory testing, a screening method that can easily detect differences in DNA sequences in addition to combinations of Ct values would be useful for rough genotyping and infection control.

As a limitation of this study, if the Ct value is high, it cannot be completely denied that the difference in amplification efficiency between the two regions is due to the detection of a low viral load, as in the case of the very early stage of infection or the recovery period, and false positives resulting from equipment or operator errors [30–33]. However, the accuracy of GX suggests that the frequency of false positives is low, as all past cases in Figure 2 were diagnosed with COVID-19. A suspected low viral load can be compensated for by retesting at a different time. Additionally, only 36 cases were selected for WGS analysis, including the cluster cases and other SARS-CoV-2-positive cases with discrepancies in Ct values and the G29179T mutation. Another possible limitation is the low sample size. Nevertheless, we were able to provide novel insights into the estimation of the infection route. Because SARS-CoV-2 will continue to mutate, quantifying this difference will not always be useful in detecting mutations. Hence, based on our observations, there is a need to establish an

assay that is capable of detecting multiple, high-frequency point mutations in the N gene region and thereby extend its framework to promote infection control.

## 5. Conclusions

In this study, we successfully identified the characteristics underlying the discrepancy in Ct values of two target genes in SARS-CoV-2 infections obtained through GX and associated these characteristics with infection control. Therefore, although not as accurate as sequencing using WGS, all information generated through qRT-PCR should be considered and could be applied to the quality control of reagents and to understand strain- and variant-specific spread by rapid screening.

**Supplementary Materials:** The following supporting information can be downloaded at https://www.mdpi.com/article/10.3390/cimb45050262/s1. Table S1. Oligonucleotide primers and probe used for analyzing the SARS-CoV-2 nucleocapsid gene. Figure S1. Comparison of the Ct delay calculated as ΔCt (N2—E) between the wild-type (G) and mutation (T) of c.29179.

**Author Contributions:** Conceptualization, K.Y. and M.M.; methodology, K.Y. and M.M.; formal analysis, K.Y. and M.M.; investigation, T.T., N.N., Y.N., A.S., A.T., S.T., J.I., O.N., K.F. and M.I.; data curation, K.Y. and M.M.; writing—original draft preparation, K.Y. and M.M.; writing—review and editing, K.Y. and M.M.; supervision, M.M. and K.Y. All authors have read and agreed to the published version of the manuscript.

**Funding:** This research received no external funding.

**Institutional Review Board Statement:** The study was conducted in accordance with the guidelines of the Declaration of Helsinki and approved by the Institutional Review Board of the Hamamatsu University School of Medicine (22-122).

**Informed Consent Statement:** Informed consent was obtained in the form of an opt-out on the website of the Hamamatsu University School of Medicine from all participants involved in this study.

**Data Availability Statement:** The data used and/or analyzed during the current study are available from the corresponding author upon reasonable request. The data are not publicly available due to privacy restrictions.

**Acknowledgments:** The authors thank all staff involved with the medical care of patients with COVID-19 and hospital infection regulations at the Hamamatsu University Hospital.

**Conflicts of Interest:** The authors declare no conflict of interest.

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
