# Peer review of "Cycle Threshold (Ct) Values of SARS-CoV-2 Detected with the GeneXpert® System and a Mutation Associated with Different Target Gene Failure"

_cimb, doi:10.3390/cimb45050262_

Round 1
Reviewer 1 Report
Comments and Suggestions for Authors
Comments:
Lines 2, 3: As other factors could affect the difference of Ct values in real-time PCR, the title could be improved as follows: “Discrepancy Between Ct Values of Different Target Genes of SARS-CoV-2 in Real-Time RT-PCR tests could be due to the Mutation in primer-binding site of one gene”.
Lines 14, 32, 73: Add detection to the Nucleic acid detection tests.
Line 25: Replace “reports” with “reports and compares”.
Line 28: The number of keywords is more than usual.
Lines 35, 36, 79, 228, 274: Use qRT-PCR instead of RT-PCR all over the manuscript.
Line 38: Add the manufacturer`s name and country of GeneXpert® System and Cepheid Xpert Xpress SARS-CoV-2 as-38 say kit (GX).
Line 49: Add “to identify the virus strains and” to clarify … to the sentence.
Line 53: As the qRT-PCR is used for quantification, the word “times” could be misleading. Use 10 cycles instead all over the manuscript.
Lines 18, 56: Replace detection site with primers and probe binding sites.
Lines 15, 61: Use real-time RT-PCR instead of real-time.
Line 75: Omit other notable symptoms. Indicate the patient`s symptoms exactly.
Line 79: Add the name of RNA extraction and cDNA synthesis kits before explaining of qRT-PCR test. If they are the same as what is used for the preparation of sequencing, move the name of RNA extraction and cDNA synthesis kits from 87-89 lines to this line.
Line 103: Add the to consensus sequence.
Line 107: Please add a statistical analysis to your study, comparing ∆ N2-E with the base of c.29179.
Line 119: What is the sample processing control (SPC)? Is it the internal control of the kit? Please explain about it in line 85 as a part of the kit.
Line 124: Replace sequence with sequencing.
Lines 169-170: Move the sentence to the line 100 as a part of material and methods.
Lines 209-210: This sentence is not true that “the difference in gene amplification efficiency—that is, the diversity of genotypes”. This sentence should be revised as follows: the diversity of genotypes is one of the most important factors affecting the difference in gene amplification efficiency. Note that there are other factors affects the difference in the efficiency of real-time PCR tests that should be rolled out. Please add a paragraph and explain about other factors that could affect the difference in the Ct values e. g. errors initiating from operator, inconsistency in reagent sampling due to non-calibrated samplers, factors affecting the reagent of one gene like non-proper use and reserve, errors of instruments, run to run variation, etc.
Line 270: The difference between Ct values is an alarm of the emergence of a new strain in the patients and a researcher cannot be sure of mutation before sequencing. So, they should add some uncertainty to their conclusion.
Comments on the Quality of English LanguageNo comment.
Reviewer 2 Report
Comments and Suggestions for Authors
The title can be modified: Cycle threshold (Ct) values of SARS-CoV-2 detected with GeneXpert® System and mutation associated with different target genes failure.
The abstract: The cycle threshold (Ct) value of the N2 region of the nucleocapsid 17 gene of SARS-CoV-2 was approximately 10 times higher than that of the envelope gene. (No cycle threshold (Ct) value approximately 10 times higher than that of the envelope gene from table 1 was seen. This is not true. The Ct should be defined in terms of number Cycles (e.g Ct of 25 versus of 35, then difference in-terms of cycle difference with be 10).
Line 40 and 41: If a sample is positive, it is expected that both targets will be detected with similar cycle threshold (Ct) values [6].
o This should be as per Xpert Xpress SARS-CoV-2 package insert interpretation of results which states: The SARS-CoV-2 signal for the N2 nucleic acid target or signals for both nucleic acid targets (N2 and E) have a Ct within the valid range and endpoint above the minimum setting. That is for positive sample
- Line 113 – 114: The Ct values (mean ± standard deviation) of positive cases with the unusual delay (17 cases) were 23.2 ± 6.6 for N2 and 34.1 ± 7.3 for E. (This results cannot be defined as 10 times more. Yes, there were higher).
- Line 153 – 154: Out of 345 SARS-CoV-2-positive cases, 15 had E2-negative, N2-positive results with high Ct values. (but still not 10 times as per the abstract
- Line 157 – 158: Ct values for E exceeded 30 in these cases, it is possible that if the Ct values deviated by approximately 10, the Ct value of N2 exceeded 45 and became undetectable.
o High Ct-values may cause interpretative difficulties. They may represent the detection of small quantities viral RNA at the beginning of an infection, or the end of an infection with persistence of viral RNA. It may also represent a false-positive result where no viral RNA is present in the sample.
Discussion:
Line 208 – 209: Since GX amplifies two genes in the viral genome and separately reports the Ct value of each respective gene, the difference in gene amplification efficiency—that is, the diversity of genotypes—can be inferred from the relationship between the Ct values of the two 210 genes. (this is not always true as these may represent the detection of small quantities viral RNA at the beginning of an infection, or the end of an infection with persistence of viral RNA)
Reviewer 3 Report
Comments and Suggestions for Authors
In this research, the authors have identified the characteristics underlying discrepancy between the Ct values of N2 and E in SARS-CoV-2 infections using GX. This study is useful for viral strain identification. This is exciting research and suitable for the journal. The following comment may help the authors to improve the manuscript before acceptance.
Can you introduce more about GeneXpert® System and Cepheid Xpert Xpress SARS-CoV-2 assay kit?
Comments on the Quality of English LanguageThe quality of English is good.
Round 2
Reviewer 2 Report
Comments and Suggestions for Authors
Revised version much improved